# Digital Food Retail: Public Health Opportunities

**DOI:** 10.3390/nu13113789

**Published:** 2021-10-26

**Authors:** Melissa Anne Fernandez, Kim Denise Raine

**Affiliations:** School of Public Health, University of Alberta, 4-308 Edmonton Clinic Health Academy, 11405-87 Ave., Edmonton, AB T6G 1C9, Canada; kim.raine@ualberta.ca

**Keywords:** digital food retail, online groceries, digital food apps, meal kits, food environments, nutrition, convenience, time

## Abstract

For over two decades, digital food retail services have been emerging alongside advances in mobile technology and improved access to wi-fi. Digitalization has driven changes within the food environment, complicating an already complex system that influences food-related behaviors and eating practices. Digital food retail services support an infrastructure that enhances commercial food systems by extending access to and availability of highly processed foods, further escalating poor dietary intakes. However, digital food retail services are heterogeneous–food delivery apps, online groceries, and meal kits–and can be feasibly adapted to nutrition interventions and personalized to individual needs. Although sparse, new evidence indicates great potential for digital food retail services to address food insecurity in urban areas and to support healthy eating by making it easier to select, plan, and prepare meals. Digital food retail services are a product of the digital transformation that reflect consumers’ constant need for convenience, which must be addressed in future research and interventions. This paper will discuss public health opportunities that are emerging from the global uptake of digital food retail services, with a focus on online groceries, food delivery apps, and meal kits.

## 1. Introduction

Consumer trends have changed rapidly along with technology, creating a market for online retail services, which extends to food [1]. Expanding e-commerce opportunities–made possible through mobile technology and widespread access to wireless Internet–are altering consumer food shopping behaviors [2]. The digitalization of the food environment [3] has generated e-commerce opportunities for food retailers to sell foods online, which are delivered or picked up offline [4]. Transformation in the food retail industry has accelerated rapidly and may provide opportunities to build resilience within the food system to stressors such as the COVID-19 pandemic [4]. Virtual points of purchase can make healthy and unhealthy foods more accessible and food purchasing more convenient [5]. A multitude of digital food retail services has sprung up, enabling food purchasing through virtual platforms (websites and apps) and offering in-person/contactless pick-up or delivery [6]. Many types of digital food retail services exist; however, this paper will focus on three prominent categories that have global reach and market penetration: online groceries (e.g., AmazonFresh) [2], food delivery apps (e.g., UberEats) [6], and meal kits (e.g., Hello Fresh) [7].

Though digital food retail has become ubiquitous in a relatively short time, it is not a new or recent concept for the food retail industry [2]. The first digital food retail services appeared in the 1990s; however, food-related e-commerce models have only recently become viable with near-universal use of the Internet and mobile technologies that allow for sophisticated digital personalized shopping experiences [1,6,8]. Companies offering restaurant delivery and online groceries emerged as early as 1994 and 1997, respectively [6,8]. Improvements in e-commerce technology and evolving consumer expectations have changed the supply and uptake of digital food retail services [8]. In earlier years, barriers to the adoption of online food retail services were related to delivery fees, internet accessibility, and the ability to offer secure mobile payment options [8]. However, today, Internet accessibility and high-speed wireless infrastructure are nearly universal, with Internet penetration exceeding 80% in Europe and North America and mobile Internet penetration exceeding 75% in most countries around the globe, allowing for seamless e-commerce experiences [9]. Online food purchasing represents a sizeable retail category with a variety of companies that extend beyond traditional grocery stores and standard pizza delivery [6,10]. These companies offer a range of competitive services that provide consumers with various aspects of convenience: saving time, decreasing the burden of meal planning and/or preparation, shopping from home, making purchases outside of store hours, and eliminating travel to brick-and-mortar stores [8,11].

The digitalization of the food environment has enabled new forms of selling and purchasing food [3] and has spawned unprecedented access to foodservice outlets, day-and-night, which has made food acquisition, healthy and unhealthy, effortless with the swipe of a finger or a few clicks. Though improved access to food may seem inherently positive, it comes with a host of challenges associated with limitless access to highly processed foods. Despite the ubiquitous use of digital food retail services globally, little health-oriented research has been conducted on online food purchasing, perhaps reflecting their relatively recent popularity and the heterogeneity of the services offered. Given the rapidly changing and dynamic nature of the online food and grocery retail sector [8], there is limited evidence to support health and nutrition policies or even guidance. Nevertheless, insight may be gleaned from emerging research and consumer reports.

Digital food retail services were burgeoning prior to the COVID-19 pandemic. Restrictions to in-person activities transformed digital food retail from an e-commerce innovation into an essential service that altered food shopping behaviors further bolstering online food purchasing [6]. The COVID-19 pandemic has merely augmented consumer intentions to adopt digital food retail, through online ordering and contactless delivery [12,13]. Though the forecasted growth of the O2O food retail sector may slow down in the future, it will continue to grow over the next five years. Ultimately, digital food retail will persist, and public health professionals need to review nutrition policies and update nutrition promotion strategies to support consumers within digitized food environments [14]. Public health opportunities and innovations can emerge from digital food retail services to support healthy eating and the needs of vulnerable populations. This review will discuss digital food retail services from a public health and nutrition perspective with a focus on online groceries, food delivery apps, and meal kits.

## 2. Key Benefits: Convenience and Time

Most online food purchasing services have been studied as business opportunities to maintain or increase the food market share, and not in relation to health or nutrition [8,15]. Early studies focused on assessing consumer characteristics, perceptions, and demand for online groceries in the U.S. [16]. While the adoption and breadth of digital food retail services have expanded over the years, the underlying contexts for purchasing food through virtual platforms are unchanged. Over 20 years ago, Morganosky and Cude reported that convenience and saving time were the most cited factors for using online groceries [16]. At the time, these factors were unsurprising, and today, they continue to resonate. In 2020, 20 years later, two of the top consumer benefits of using online food delivery remain convenience and saving time [6,17,18]. Other benefits include effortless ordering, the ability to make informed choices, discovering new cuisines, greater selection, and the ability to make choices based on patron reviews [6]. Convenience and time are intertwined with contemporary lifestyles and convenience foods and services respond to a particular but ubiquitous need to acquire and prepare food with minimal effort [19]. The historical context for these needs can be linked back to the transition of women into the workforce, the automation of food production, a loss of cooking skills, and an increase in household disposable income [20]. Digital food retail is merely a practical innovation that fills consumers’ needs for both convenience and time.

Perceived time scarcity is not only a common reason for using digital food retail services but is also a reason for consuming highly processed foods and can be a substantial barrier to healthy eating [20,21,22,23,24]. Food preparation involves more than just mechanical cooking skills (e.g., chopping, mixing, heating), but rather encompasses multiple complex behaviors: planning (budgeting, selecting recipes, meal planning), cooking, food storage, and kitchen cleanup [25,26]. Convenience foods significantly reduce the effort and time involved in food preparation. Similarly, digital food retail services will reduce the burden of food preparation; however, the impacts depend on the type of service used. For example, digital food apps replace nearly all the steps involved in food preparation, while meal kits only replace planning steps, and online groceries replace the time spent traveling to and from brick-and-mortar stores or physically searching for items. Whether real or perceived, households that use digital food retail services have a legitimate belief that they are saving time by using the services.

Public health professionals must not underestimate the power of consumers’ need for convenient and timely food preparation. Leveraging existing innovations within the digital food retail sector can be a valuable strategy to support healthy diets in ways that respond to longstanding consumer needs–convenience and saving time. E-commerce and O2O business models have advanced the web-based technology, order/delivery infrastructure, and consumer experience to a degree that can be adapted to fit with public health objectives in most urban and suburban dwelling areas worldwide. Introducing health aspects as an added value to already existing or new digital food retail services is not only feasible but timely.

## 3. Online Groceries

In 2000, Morganosky et al. observed that populations who may find grocery shopping physically or logistically challenging (e.g., parents, older adults, and individuals with limited mobility or health conditions) could benefit considerably from online groceries, packing, and/or delivery services [16]. Furthermore, by 2004, literature began to suggest that online groceries could be tested as a strategy to improve access to fresh fruits and vegetables for groups with low intakes [27]. Researchers have indeed been studying innovative interventions that leverage online groceries to improve eating decisions for over a decade. Yet, relatively little evidence exists on the types of interventions that could be implemented effectively in digital food retail environments. There is even less evidence on the public health policy implications of food retail environments on diet and health, let alone digital food retail environments. Mah et al. (2019) highlight that public policy research on (brick-and-mortar) food retail focuses on the consumer and misses opportunities to intervene in the broader food environment by creating interventions that include retailers and suppliers [28]. Grocery stores can play a pivotal role in public health, positioned between food marketers and consumer food purchases [28,29]. Unlike other digital food retail services, online groceries will not substantially reduce the food preparation burden (i.e., time spent cooking). However, interventions conducted in grocery stores have been effective in promoting the purchase of healthy foods [30]. Leveraging the marketing mix (the 4 Ps, products, price, placement, and promotion) to promote health is more feasible in an online setting than at a physical grocery store. By extrapolating evidence from in-store marketing studies on influencing food-purchasing behaviors we highlight promising strategies and research needs to promote healthy eating through online groceries (Table 1) [29]. Online groceries are an innovation that fit the needs of populations of interest (e.g., older adults) and could be easily adapted to nutrition education and promotion interventions at the point of purchase. Consumers are interested in services that are convenient, timely, and provide added value like health features. There are opportunities for retailers to optimize user experiences, retain, and attract patrons while supporting healthy eating by implementing added features to online platforms: shopping cart rating tool, healthy meal planning tool, settings for healthy shopping [31].

Digital food retail services provide cost-effective platforms to implement behavioral interventions at the point of purchase [32]. Virtual platforms are particularly appropriate for nudging since they are automated and can be personalized to consumer preferences throughout the shopping experience. Nutrition education and nudging interventions on digital food retail platforms have been successfully demonstrated in a handful of studies. In one study, using an Internet shopping site, participants in the effect group were given real-time recommendations to select products lower in saturated fat while participants in the control group received no recommendations. The intervention group had significantly lower amounts of saturated fat in their online grocery basket compared to the control group. Additionally, the quantity of food purchased and dollars spent did not differ between groups [32]. Interventions that provide personalized dietary advice while online grocery shopping are easy to implement, sustainable to maintain, and can alter food purchasing behaviors. Gorin et al. examined the impact and acceptability of using online groceries for a behavioral weight loss intervention [33]. The study rationale was based on modifying participants’ household food environments through food shopping rather than merely providing a calculated amount of food to the participant enrolled in the study. Subjects in the 8-week weight loss intervention were randomized to receive either a standard weight-loss treatment or an online grocery delivery + standard weight loss treatment. By the end of the intervention, there were no differences in weight loss between the groups; however, the online grocery group purchased fewer high-fat foods for their households. Furthermore, participants in the online groceries group reported that online grocery shopping helped decrease impulse purchasing and supported the selection of healthier foods. The study suggested that ordering groceries online could reduce unhealthy eating triggers within household food environments [33]. In addition to modifying the household food environment, individual health concerns can be addressed on online grocery platforms. For example, in an online simulated grocery shopping experiment, researchers examined whether fiber-focused prompts at the point-of-decision elicited participants to choose fiber-rich foods. Participants who received point-of-decision prompts for fiber chose products higher in fiber and with higher healthiness ratings compared with participants who did not receive any prompts. Findings from this study support the use of point-of-purchase nudging on online grocery platforms to stimulate healthy food purchases [34].

The notion that online groceries dissuade impulse purchases has been cited in multiple studies since 2000 [16,33,35,36] and continues to be explored [17,37]. Recently, a qualitative study observed that impulse purchases were greater online than in person; however, participants reported that temptation was greater in-store, and impulse purchases in-store tended to be unhealthy foods [17]. Buying groceries on virtual platforms can reduce total food dollars spent and the quantity of unhealthy foods purchased [17,18,38]. Generally, consumers are more likely to spend more money on items when benefits are immediate and tangible (e.g., in-store purchases) and are more likely to purchase items with short-term benefits (e.g., tasty foods high in sugar, salt, fat) rather than long-term benefits (e.g., healthy foods). The temporal delay between completing an order online and receiving the order offline can alter purchasing behaviors, interrupting immediate impulses to purchase highly processed foods [35]. Online groceries provide features that support meal planning [18]. These planning behaviors guide the purchase of foods and ingredients that are intended for specific meals and/or recipes, reducing the impulse to purchase miscellaneous foods (e.g., ice cream) [35]. The visual display of foods virtually on a screen reduces its vividness, which curbs consumers’ motivation to purchase foods that provide instant gratification and further minimizes impulse purchasing [36]. The temporal delay between placing an order and receiving it, planning behaviors, and reduced visual stimulation online all contribute to reducing impulse buying among online grocery shoppers. Reduced impulse purchasing of discretionary foods is an understated benefit of online grocery services that could be featured by public health practitioners to improve food purchasing behaviors and potentially save money with little effort. Further combining meal planning behaviors with online grocery shopping will maximize opportunities to improve household food environments.

In addition to behavioral interventions that promote healthy food purchasing, online grocery services could supply healthy food to areas with poor access (food availability or affordability) and address transportation barriers [39]. Policymakers and public health practitioners have touted online groceries as a solution to food access challenges, particularly among groups with a low income, limited mobility, older adults, and individuals without vehicles. As part of a program to support independent living among older adults, the government of the province of Quebec in Canada provides a tax credit to reimburse older adults fees related to grocery cart assembly, delivery, and tipping [40]. In Montreal, Quebec, a study found that the loss of physical or motorized mobility among older adults was associated with greater intentions to use online grocery services. This study suggests that digital food retail services are a promising opportunity to improve food access among older adults living in urban areas [41]. In the U.S., an initial pilot study confirmed that online grocery services have the potential to improve food access, particularly when optimized with competitive pricing, quick affordable delivery, and accept food assistance vouchers [42]. A Virtual Super Market was successfully tested in food deserts in Baltimore to improve access to healthy foods. Online grocery orders were placed and delivered to neighborhood hubs that were within walking distance (e.g., library, school, or senior housing center). The pilot demonstrated that online food shopping was feasible, acceptable, and participants favorably reported that the program increased the availability of healthy food and reduced the need for transportation to and from the market [39].

In 2014, the U.S. Department of Agriculture (USDA) Farm Bill included provisions to pilot Supplemental Nutrition Assistance Program (SNAP) benefits to pay for groceries online as a potential solution to neighborhoods with limited access to healthy foods and grocery stores [39]. A pilot program launched in 2019 in New York authorized SNAP participants to redeem their benefits online at popular retailers such as ALDI, Amazon, and Walmart and has now been expanded across the U.S. [43]. In the initial Online Purchase Pilot, SNAP benefits were accepted online in eight states. A study examining the availability of online grocery delivery within these states reported that only 5.9% of census tracts containing food deserts in urban neighborhoods did not have any online grocery available. Conversely, in rural neighborhoods, 69.5% of census tracts that had food deserts did not have any access to online groceries, and the remaining 30.5% of census tracts only had access to services that were partially deliverable. Results from this study indicate that there is excellent accessibility to online groceries for SNAP households in urban areas. However, dismal access to online groceries in rural areas will limit the relevance of the online grocery programs to urban households [44]. During the COVID-19 pandemic, preliminary data showed that a greater proportion of SNAP participants began using online groceries compared to individuals without SNAP benefits. However, participants of Women, Infants, and Children (WIC) were not able to order groceries online, despite high interest. 38% of SNAP participants also reported not being able to buy groceries online, in part because some stores do not accept SNAP benefits or delivery windows were too far in the future [13]. Although current programs are not inclusive of all vulnerable populations, online grocers can become powerful allies in improving food insecurity. As new data emerge, food assistance programs can be further adapted to target rural neighborhoods, WIC participants, or other vulnerable groups. In a poverty-targeted intervention for pediatric cancer, groceries were provided to low-income families through an online service, Instacart. Participants reported that Instacart gift cards allowed them to purchase foods perceived to be healthier and allowed them to cater to the preferences of children during chemotherapy. However, there was still hesitancy to purchase fresh foods like meat and produce [45]. Despite the national roll-out of SNAP benefits at online retailers, there is little evidence as to whether the services influence the purchase of healthy food. In addition to improving household food environments, online groceries offer valuable social benefits: reduce financial stress, improve equitable access to food, foster community, and support families during stressful times [13,39,45].

The inability to physically examine products has been a major challenge limiting the adoption of online groceries [10]. Consumers prefer to select fresh products like produce themselves, and it is impractical and less likely for them to examine food labels and nutrition information when making purchases online [10,46]. Other substantial barriers that prevent consumers from adopting online food shopping include fees, concerns about the quality of perishable items, and a lack of trust in the online shopping process [46]. For these reasons, previous research has suggested that SNAP participants lacked interest in purchasing groceries online [46,47]. The COVID-19 pandemic has since normalized online grocery shopping, increasing consumer adoption and prompting new intentions to use online grocery services [48]. Older adults are particularly hesitant to use new technology and are suspicious of online payment. Older adults are more likely to sustain the use of online groceries after a successful trial [41]. Helping older adults with tasks such as creating an online account or filling virtual grocery baskets can build confidence to use digital food retail services independently [39,49]. Digital food retail services are tools to improve access to food but may not be sufficient to alter dietary diversity or improve dietary intakes among older adults [50].

There is evidence that online shoppers seek familiar items by focusing on pictures and are less likely to read detailed product information [51]. Brand loyalty is higher online compared to offline when food shopping [52], potentially indicating that consumers may rely on previous knowledge to select products. Given that consumers are more likely to look at product pictures rather than examine detailed product information [51], marketing strategies such as packaging or health claims and health education strategies such as front-of-package-labeling or the nutrition facts table may have limited utility for online shoppers. Nutrition information for packaged products is not always available online [53], and when it is available, key information is presented inconsistently [54]. Available information may be inaccurate or hard to view without additional scrolling, zooming in, or clicking. Furthermore, all products can be purchased without viewing any nutritional information [55]. Online platforms provide opportunities for food companies to promote products (through ads and pop-ups), but also for public health practitioners to work with retail companies to nudge consumers to make healthier choices by suggesting product swaps, programing reminders to purchase healthy foods, or pre-filling carts with default healthy items [56,57,58]. However, online groceries may also increase consumption of convenience and highly processed foods by making them more accessible [38]. To maximize the purchase of healthy foods and beverages, online grocery food access programs ought to be complemented by interventions that reduce discretionary food or incentivize healthy food purchases [42].

## 4. Food Delivery Apps

The use of food delivery apps continues to rise, and global revenue is projected to increase by an average of 12.6% a year for the next five years [6]. Food delivery apps are the main form of online food delivery products and platforms offering either restaurant-to-consumer delivery and platform-to-consumer delivery. Despite the common mention of “delivery”, both of these business models have options for customers to pick up orders in the restaurant as well [6]. Food delivery apps are merely web-based platforms that are generally accessed on mobile devices but can also be used on typical browsers on any device without downloading an app. The term ‘food delivery app’ is used interchangeably with ‘online food delivery’ or ‘online takeaway’ and has become synonymous with the general term ‘food delivery’. In this review, we will use the term ‘food delivery app’ and focus on the platform-to-consumer model, whereby companies such as Uber Eats act as an intermediary between foodservice outlets and customers. In this model, customers order prepared meals, snacks, or beverages online from a third-party service that are then prepared by the food service establishment, packaged, and couriered to the customer. These services all have the ability to track information about the order confirmation, preparation, payment, and delivery that can be shared with the customer in real-time [6]. Order information and data tracking make services seamless and easy to use, attributes that are highly sought by consumers. The information collected can also be used to improve customer experience, provide personalized promotions, and create recurrent orders. From a research perspective, this data can be extremely valuable in helping public health practitioners understand consumer food purchasing behavior, providing insights into the frequency and quality of foods consumed outside of the home. Furthermore, a host of information about restaurants, location, and menu items is available and can be harnessed with a little effort to provide valuable data on the nutrition quality of extensive food offerings to monitor the out-of-home food environment [14].

Regardless of the type of service or their specific attributes, food delivery apps respond to an ever-increasing consumer demand for convenience [59] and can make accessing the food environment easier by removing physical barriers to acquiring food–both healthy and unhealthy [60]. For example, traditional neighborhood food environments are generally limited to a 1.6 km-radius or a 20-min walk from home, school, or work, whereas food delivery apps substantially expand the neighborhood food environment by up to a 10 km-radius [61]. The neighborhood food environment could be further expanded through innovations that aim to make foodservice more convenient. Companies are already testing self-driving cars, drones, and delivery robots [6]. For example, in South Korea, the feasibility and acceptability of using drones for food delivery are being investigated [62].

Though food delivery apps offer a wide range of foods, the items purchased, like most out-of-home meals, are likely large portions of foods that contain excess saturated fat, sugar, and sodium [63]. 87% of the largest restaurant brands (e.g., McDonald’s, Wendy’s, and Chipotle) in the U.S. have partnerships with DoorDash, the biggest online food delivery company in the U.S. These companies actively promote highly processed foods on food delivery apps [64]. Market research provides insight into the types of foods that are most commonly ordered. For example, in the UK, the top 9 foods ordered were pizza, burger, salad, fish and chips, pasta, curry, sandwich, kebab, and fried chicken [6]. Although food delivery apps offer healthy foods, the foods purchased are predominantly highly processed. Initial research out of Australia and New Zealand confirms that the majority of the most popular food service establishments operating on the leading food delivery app platform, UberEats, serve unhealthy foods, and that the most popular foods are unhealthy [65]. High consumption of out-of-home meals has been associated with higher BMI and cardiometabolic risk factors [66]. Researchers in the UK have demonstrated the feasibility of developing a health rating score that can be applied to foodservice establishment listings on food delivery apps; however, it is not known whether the score would be deemed valuable for consumers or induce healthier choices [67]. Although there is limited knowledge about the impacts of food delivery apps on nutrition or health, there are concerns that normalized use will stimulate the consumption of out-of-home foods [68], and by extension, a rise in the consumption of nutrient-poor energy-dense foods. We assume that meals purchased through food delivery apps are being consumed in addition to other highly processed meals; however, they may simply be replacing the consumption of existing highly processed ready-to-heat foods (e.g., frozen pizza) or an in-person meal at a restaurant (e.g., a local pub).

Food delivery apps are an expensive service [6] and healthier food options like salad are even more expensive when purchased through an app. Increased accessibility of processed foods through food delivery apps could disproportionately impact individuals with a lower socioeconomic status, who may not be able to afford to select healthier options from an already expensive service. A multi-country study including Australia, Canada, Mexico, the UK, and the U.S. on online food delivery found that users were more likely to be male, identify with a minority, be highly educated, and live in a household with children [68]. Market research illustrates similar trends with a higher proportion of male users. In addition, we note similar proportions of users from low-, middle-, and high-income groups across countries ranging from 30–41%, 30–39%, and 27–37%, respectively (Figure 1). In the U.S. and Spain, there is a greater proportion of high-income food delivery app users than low-income users, whereas the opposite is true for China, the UK, Germany, and Italy. Though differences in the proportion of users from low- and high-income groups exist between countries, overall, food delivery apps are highly accessed by all groups, regardless of income (Figure 1). The question that remains is whether the quality of foods purchased through food delivery apps differs between income groups.

Young adults are the age group with the highest proportion of food retail app use [6,68,69] and make independent eating and spending decisions, rendering them a population of interest. Though sparse evidence is available, the effects of high online food delivery use likely extend beyond nutritional risk factors to include mental health, similar to offline out-of-home meal consumption [70]. One study among Chinese college students reported that frequent consumption of online takeaway was associated with a higher risk of emotional overeating [69]. Aside from poor nutrition and mental health outcomes, the increased use of food delivery apps has broader public health implications for physical inactivity and excessive waste [14]. Researchers in China have suggested that the convenience of delivery services may contribute to a sedentary lifestyle by decreasing physical activity as individuals no longer need to leave their home, school, or workplace to obtain a meal [61]. Packaging waste from food delivery services is becoming a major environmental concern in cities where food delivery services are highly accessed [71,72]. Additionally, order minimums for free delivery, discounts, and lucrative promotions are industry tactics to incentivize over-ordering, which hypothetically contributes to food waste [73].

Online food delivery business models, order processing, and delivery logistics are currently being leveraged and adapted by community groups and the charitable food sector to provide local food assistance. It would be valuable to investigate the scalability of successful programs such as Meals on Wheels that have changed over time to provide online food ordering and local delivery of ready-to-heat meals customized to dietary restrictions and are available to all community members regardless of age, income, or health [74]. The digitalization of many charitable food programs at scale was precipitated by the COVID-19 pandemic, which disrupted traditional food banks and spurred greater food insecurity. Innovations during this time included programs going online and creating temporary partnerships with food delivery giants or entirely new food delivery models. The economic viability of many food-assistance programs relies on unpaid labor (i.e., volunteers), and further digitalization of these services could support programming, making them more efficient, ultimately reducing the number of unpaid hours needed to function [75]. Tailored digital innovations can effectively support the charitable food sector, particularly during times of crisis (e.g., COVID-19 pandemic); however, they have always been mere Band-Aids in addressing chronic food insecurity and poverty [76].

## 5. Meal Kits

Meal kits, the newest online food shopping innovation to gain popularity, offer fresh packages of raw ingredients with cooking instructions that save time and remove the burden of meal planning [77]. Meal kit providers may also offer high-quality ready-to-heat meals of meals that require simple assembly with little-to-no preparation [78]. Meal kits make cooking simpler and appeal to time-poor consumers [79]; however, they are more expensive than cooking a meal with self-selected recipes and ingredients [80]. Given the increased popularity of meal kits, big food has taken an interest, with companies like Campbell Soup and Nestlé backing the largest international meal kit services [81]. Compared to food delivery apps, we speculate that meal kits likely provide smaller portions and greater amounts of vegetables, potentially making it easier to eat healthier. The average cost of a meal from popular meal kit providers in the U.S. ranged from USD 6.99 to 10.99 per portion in 2019 [7], which is more affordable than eating out. Finally, compared to online groceries, meal kits remove barriers to home cooking by providing instructions, recipes, and pre-portioned semi-prepped foods [78]. Among digital food retail services, meal kits have generated the least evidence, leaving us to rely on speculation about the impact that meal kits may have on diet quality and cooking practices. Sparse evidence does, however, provide some insight.

Two separate Australian studies published in 2019 and 2020, respectively, found that meal kits provided a sufficient amount of vegetables but could be high in sodium, and macronutrient content varied considerably from recipe to recipe [82,83]. Results from both studies suggest that opportunities exist to improve the nutritional quality of meal kits, alter recipe directions to align better with dietary guidelines, and provide recommendations to modify recipes tailored to dietary concerns [82]. In an American study published in 2021, college students were randomized to one of four groups to study food agency (ability to plan, procure, and prepare food), cooking frequency, and diet quality: (1) cooking classes (2) cooking classes + meal kit provision, (3) meal kit provision, and (4) no intervention. The authors found that although meal kits improved food agency, cooking classes designed to improve food agency were more effective. Additionally, meal kits did not have any significant effects on cooking frequency or diet quality [84]. Though meal kits are typically marketed towards higher-income households, there is evidence that meal kit interventions would be acceptable and highly utilized among families with low incomes. However, the number of low-income families willing to pay for a weekly meal kit service is generally less than the price of meal kits sold by major national companies [85]. Danish studies have suggested that meal kits embody contemporary food preparation. Meal kits combine a lower threshold of mechanical cooking skills with digital literacy to enable households to prepare home-cooked meals without needing substantial food literacy [86]. Meal kits can remove the food preparation burden (planning and shopping), making it easier to actually cook meals at home and prepare foods that fit with dietary preferences, restrictions, or health concerns [87]. However, it remains unclear whether meal kits foster sustained cooking skill development or enduring food agency.

Cooking and home food preparation are determinants of healthy eating [23]; however, cooking skills and cooking frequency alone are insufficient to improve diet quality [88]. Cooking skills need to be combined with food and nutrition knowledge, including how to select healthy foods and then prepare them (food agency) [89]. Meal kits can support self-taught cooking and increase cooking confidence but once the supports of the meal kits are removed, it is not clear whether individuals will have the agency to continue to cook meals with minimally processed ingredients. We speculate that the impacts of meal kits on both cooking practices and diet quality are short-lived. Nevertheless, meal kits afford stress relief to overwhelmed individuals by reducing the burden of meal preparation (planning recipes and shopping for ingredients). Unlike other digital food retail services, the COVID-19 pandemic is not expected to significantly boost the meal kit market, as a small percentage of individuals intend to continue using meal kits after the pandemic [7]. The benefits of meal kits are likely limited to groups with higher incomes who have greater intentions to continue subscriptions [7]. For these reasons, interventions developed to fit with conventional meal kits are likely to have limited impact.

## 6. Key Challenges: Evolving Technology and Complex Commercial Food Systems

The cost of a computer or mobile phone with Internet is one of the main barriers to not using the Internet [9]. Limited internet connectivity may impact older adults who are not comfortable with technology and do not own a computer/mobile phone, individuals who cannot afford Internet provider fees, or simply populations who live outside city zones with reliable Internet access. Additionally, adults over the age of 55 are less likely to order online groceries [13], rural residents lack access [44], and lower-income groups have limited willingness to pay for expensive meal kits [7,85]. We need to be aware that as digital food retail services expand, they may be contributing to the digital divide by providing excellent opportunities for improved food access to groups who live in wealthy, well-connected neighborhoods but limited options for groups with access to fewer resources. Similarly, any interventions that focus on digital food retail may entirely miss key vulnerable populations who lack simple Internet access. There are legitimate concerns that the digitalization of health and food systems could broaden gaps between socio-economic groups in favor of already privileged consumers [90].

As consumer data and tracking information is collected, the digital food retail sector will continue to pivot to meet customer demands for seamless services that grant multiple ways to order, purchase, and obtain food, all while maximizing revenue [91]. We can expect user interfaces to continue evolving with new tactics that will target consumers with greater precision to increase order size and frequency, and ultimately, dollars spent [91,92]. Digital food retail environments are part of a complex commercial food system that fosters high-volume sales of highly processed foods to achieve profit [93]. Within this commercial food system, digital food retail environments have demonstrated high adaptability to stressors. Examples of this adaptability can be found throughout the COVID-19 pandemic, whereby digitalization made it possible for small and large food retailers to continue to conduct business despite severe disruptions to traditional in-person operations [4]. Our public health system, however, is not as adaptable or reactive.

The fundamental public health challenge within the digital food environment is to develop solutions (programs and policies) within a constantly evolving commercial food system. Adding to this challenge, digital food environments are not fully understood, making it a blurry target to address. We simply do not know the extent to which the digitalization of the food environment will impact consumer food purchasing behaviors and thereby health. Thus, we are left with substantial knowledge gaps regarding the impacts of the ongoing uptake and normalized use of digital food retail services on diet-related health outcomes [94]. However, we can speculate that further commercialization of the food environment will augment the consumption of highly processed foods, promoting diet-related diseases, and will expand equity gaps if left unchecked. Furthermore, we can speculate that there will be unforeseen consequences of digital food retail services that extend beyond nutrition and into waste management, and threaten environmental protection, workers’ safety and rights, and data stewardship. We are indeed working on complex problems (nutrition and health) within an increasingly complex commercial food system. Solutions need to match the complexity of the problems while recognizing the complexity of addressing dynamic digital food environments. Finding common ground between commercial food systems and public health policy is a first step to tackle the evolving digital food environment. Immediate actions that align commercial food systems with public health policy include working with food delivery apps and online grocers to promote the sales of healthier foods and meals [93]. Steps forward will require strong government leadership and sustained goodwill from the retail food industry.

## 7. Conclusions

The digitalization of food environments [3] generated virtual points of purchase, enabling digital food retailers (online groceries, food delivery apps, and meal kits) to spread their reach as e-commerce penetrates further into global markets [6,7,95,96]. The expansion of digital food retail services enhances commercial food systems, making it easier to sell food–healthy and unhealthy. Bridging knowledge about consumer food acquisition (i.e., shopping for food), preparation, and consumption behaviors with health is fundamental to improving diet quality within an increasingly complex food system. The foods acquired and consumed are highly influenced by the availability and affordability of retail food environments. Adopting a consumer-oriented approach in research and public health can provide new insights on barriers to healthy eating and novel solutions to build resilience within commercial food systems [4]. Digital food retail environments provide stakeholders new tools to address food insecurity and malnutrition [97]. Building on knowledge from traditional food environments, it is possible to design, test, and work towards innovations that leverage digital food retail services. Based on past research and evidence, we speculate that online groceries have the greatest potential to support healthy eating, while food delivery apps can become a substantial barrier to healthy eating. Though meal kits can also support healthy eating, the current services do not have the same reach as other digital food retail categories and primarily service higher income groups [78]. More research is needed to understand how to leverage existing services to better serve vulnerable populations and groups who typically find time scarcity a major barrier to healthy eating.

Exploring the influence that online food purchasing has on meal choices, eating practices, and diet quality is a necessary step to deepen our understanding of digital food retail environments. Online food purchasing services are likely to have an impact on the nutritional health of individuals who utilize these services regularly through mechanisms that are linked to changes in home-based food preparation, eating practices, and diet quality. Understanding what matters to consumers is key to developing appropriate interventions that involve emerging digital food retail services. Given the rapid evolution of online food purchasing and the massive knowledge gap about the impacts of new digital food retail environments on eating practices, consumer behavior and policy research are urgently needed.

## Figures and Tables

**Figure 1 nutrients-13-03789-f001:**
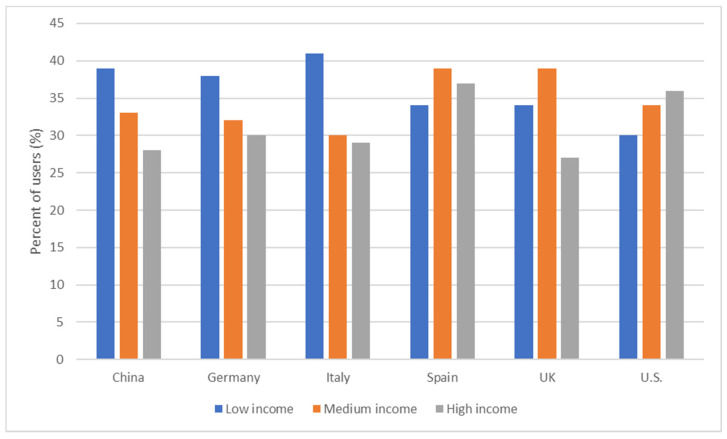
Proportions of online food delivery users from low-, middle, and high-income groups across six countries. Source: Online food delivery report [6].

**Table 1 nutrients-13-03789-t001:** Strategies to support health and nutrition by applying the marketing mix (the 4 Ps) to online grocery store environments.

*The 4 Ps*	*Strategies to Support Healthy Eating*
**Products (access)**	Ensure food deserts and food insecure neighborhoods are well serviced by online grocery stores Assist vulnerable groups with placing and receiving orders
**Product (features)**	Personalized app settings for healthy shopping Personalized meal planning and recipe tools
**Price**	Subsidize or waive delivery fees for vulnerable groups Subsidize healthier foods
**Placement**	Make nutrition information easier to view and more accurate Make minimally processed foods easy to find and highly processed foods less visible
**Promotion**	Block ads for nutrient-poor energy-dense foods
	Generate reminders/nudges to purchase minimally processed foods

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
