# Peer review of "Digital Food Retail: Public Health Opportunities"

_nutrients, 2021, doi:10.3390/nu13113789_

Round 1

Reviewer 1 Report

  • The manuscript entitled “Digital food retail: Public health opportunities” Authors developed a literature review and detected three areas with opportunities in food retail services: food delivery apps, online groceries, and meal kits. The manuscript is clearly written; this reviewer has the followings concerns:
  • L26-31. Please look at a citation according to justify each approach.  
  • L139. I am afraid I disagree that the term “manipulating” in marketing science is not used. 
  • L342. MacDonalds?
  • Is there a reason for introducing the key benefits “convenience and time”. Maybe time is within convenience. Please look at food values by Lusk.
  • Please add more explanation to present the areas food delivery apps, online groceries, and meal kits. Why?
  • Please complement the conclusion section with the past changes in the document. 

Author Response

The manuscript entitled “Digital food retail: Public health opportunities” Authors developed a literature review and detected three areas with opportunities in food retail services: food delivery apps, online groceries, and meal kits. The manuscript is clearly written; this reviewer has the followings concerns:

Thank you for providing constructive feedback

L26-31. Please look at a citation according to justify each approach.  

Citations have been added and sentences have been reworded to reflect new references.

L139. I am afraid I disagree that the term “manipulating” in marketing science is not used. 

Manipulate has been replaced with leverage

L342. MacDonalds?

Corrected to McDonalds

Is there a reason for introducing the key benefits “convenience and time”. Maybe time is within convenience. Please look at food values by Lusk.

In consumer reports and research on digital food retail purchasing services, convenience and time are consistently and separately cited as top reasons that consumers use digital retail rather than brick-and-mortar food purchasing options. Time is indeed a facet of convenience, but we believe it warrants a separate emphasis. Perceived lack of time is a commonly reported reason that consumers do not cook, feel they cannot eat healthy, use highly processed convenience foods, and use convenience services like food delivery apps or online groceries. Thank you for sharing the food values by Lusk. We acknowledge that in the study by Lusk and Briggeman (2009), convenience was rated an intermediary attribute for food selection. However, we believe the context of choosing between two different methods of shopping for food is different than choosing between two individual foods to consume.

Please add more explanation to present the areas food delivery apps, online groceries, and meal kits. Why?

These groups have been chosen because they have global reach and market penetration. These details have been added at line 38 with accompanying references for relevant consumer reports.

Please complement the conclusion section with the past changes in the document. 

Minor edits and additional references to mirror the introduction have been made in the conclusion.

Reviewer 2 Report

This is a well-written piece. 

I do not shop for food online - but is it really convenient? It takes time to find what you want to buy and order it. 

The internet is still patchy and fails, which limits using it for a big shop! Not everyone has the internet or wants to use it!

The articles states that people have 'increased incomes' that allow them to shop online. However, there are many people who live on the breadline and in fact are reliant on food banks simply to eat, so online shopping is impossible for them. 

Many people like to see what they are buying and like to cook from scratch rather than buying ready meals or processed foods.

There is no instant gratification if you buy online - it is not like going to a shop and buying a snack to eat afterwards!

It would be a great place for promotion of healthy foods/meals but there is a risk that large companies would pay to use this to advertise unhealthy foods. 

Author Response

This is a well-written piece. 

Thank you!

I do not shop for food online - but is it really convenient? It takes time to find what you want to buy and order it. 

Convenience has multiple facets. In terms of time, it is true, online groceries do not necessarily save time compared to in-store shopping. However, online grocery shopping can be done while commuting, outside regular store hours, and food items can be added intermittently to a virtual shopping cart while doing other things. Additionally, once a virtual grocery cart is purchased, the platform saves the items making them easier and quicker to find the following visit.

The internet is still patchy and fails, which limits using it for a big shop! Not everyone has the internet or wants to use it!

You are correct; however, many of these services will be limited to major urban areas globally and suburban areas in the high-income countries, where internet penetration is high and relatively reliable. In areas where the Internet quality is insufficient for e-commerce transactions, digital food retail services are unlikely to be offered. Despite some regions lagging behind with quality Internet connectivity, e-commerce continues to expand and food delivery apps are one of the fastest growing sectors of e-commerce.

The articles states that people have 'increased incomes' that allow them to shop online. However, there are many people who live on the breadline and in fact are reliant on food banks simply to eat, so online shopping is impossible for them. 

We agree, e-commerce would be impossible and impractical for some marginalized groups. However, supply and delivery logistics from digital food retail platforms can be leveraged to support the charitable food sector in some regions. Small and medium sized entrepreneurs can also leverage digital food retail platforms to reach consumers.  Online shopping will definitely benefit some groups, while others will be left behind. This is the reality of the digital divide, which is an important public health concern.

Many people like to see what they are buying and like to cook from scratch rather than buying ready meals or processed foods.

We agree. Digital food retail includes services like meal kits, which are purchased online and delivered to consumer’s households.

There is no instant gratification if you buy online - it is not like going to a shop and buying a snack to eat afterwards!

We agree, which is one of the reasons that shopping online for groceries is though to curb impulse purchases of junk food.

It would be a great place for promotion of healthy foods/meals but there is a risk that large companies would pay to use this to advertise unhealthy foods. 

Digital food retail platforms are already promoting unhealthy foods through ads, pop-ups, and featured items. We believe, however, that the platforms could also be leveraged to promote healthy foods purchasing.